# Effect of Late $^{15}$N-Fertilization and Water Deficit on Allocation into the Gluten of German and Mediterranean Spring Wheat Cultivars

Klaus-Peter Götz [1],* and Osman Erekul [2]

1   Agricultural Climatology, Faculty of Life Sciences, Humboldt-University of Berlin, Albrecht-Thaer-Weg 5, 14195 Berlin, Germany
2   Department of Crop Science, Faculty of Agriculture, Aydin Adnan Menderes University, 09100 Aydin, Turkey
*   Correspondence: klaus-peter.goetz@agrar.hu-berlin.de; Tel.: +49-(0)30-2093-46390

**Abstract:** In a split N-application system, the objective was to quantify N/$^{15}$N in gluten and non-gluten proteins after the late application of 30 or 60 kg N, whereby 10% of the third split was applied as $^{15}$N. This fertilization was combined with a reduced water supply for 21 days (well-watered (ww); water deficit (wd)). German spring wheat cultivars, Elite wheat Taifun, Quality wheat Monsun and cultivars from the Mediterranean territory, Golia, Gönen, were examined. The protein content in gluten was for 30 kg N, ww, similar for Taifun, Golia, and Gönen, but markedly lower in Monsun (231, 245, 247, 194 mg protein/g DM). The water deficit increased the protein content in the gluten of Golia and Gönen and was higher than that of Taifun and Monsun (297, 257, 249, 202 mg protein/g DM). Fertilization of 60 kg N, ww, did not result in any change in the protein content in gluten and differences between the cultivars were not detectable. The $^{15}$N protein in gluten was for 30 kg N, ww, markedly higher in Gönen (2.32 mg $^{15}$N protein/g DM), compared to Golia and Monsun (1.93, 1.50 mg $^{15}$N protein/g DM), and similar in Taifun (1.64 mg $^{15}$N protein/g DM). $^{15}$N fertilizer uptake into gluten was stimulated by water deficit for 30 and 60 kg N, leading to significantly increased $^{15}$N protein in Golia and Gönen, (2.38, 2.99, 4.34, 5.87 mg $^{15}$N protein/g DM). Fertilization of 60 kg N led to a proportional two-time increase in the $^{15}$N gluten protein of the four cultivars, in ww and wd plants. Assessed on the basis of $^{15}$N fertilizer allocation under wd conditions into gluten proteins, Golia and Gönen have a stronger sink activity, compared to Taifun and Monsun.

**Keywords:** $^{15}$N dilution technique; spring wheat; gluten; non-gluten; N fertilization; internal quality

## 1. Introduction

Wheat yield and quality traits are highly influenced by the interplay of environmental factors, such as water supply, air temperature and agronomic management practices, as selection of suitable cultivars for a certain location and nitrogen (N) fertilization [1]. High N application geared to the needs of the plants during the different phenological phases can increase wheat yield and its quality; indeed, excessive use of N fertilizers leads to environmental load. Significant progress has been achieved in recent decades through breeding directed towards improved grain yield and baking quality, but the influence of breeding during the last century on wheat protein contents is inconclusive, reporting an increase, no change, or a decrease [2]. The amount of N fertilizer applied and the timing of its application strongly increases grain quality. Generally, application of high amounts of N fertilizer increases storage protein content, endosperm protein body quantity, and flour-processing quality by altering the expression levels of genes that encode components of protein biosynthesis pathways [3]. The crude protein content in grain is the main quality criterion for wheat, especially for bread making. The different wheat proteins are traditionally classified into so-called Osborne fractions according to their solubility. Gliadins are soluble in aqueous alcohols (e.g., 60–70% ethanol). Glutenins are only partly soluble

in diluted acids and bases or completely soluble in alcoholic solvents containing disaggregating and reducing agents. Gliadins and glutenins are storage proteins and account for 75–80% of wheat proteins and form the viscoelastic mass known as gluten. Albumins and globulins are soluble in water and salt solution. The albumin/globulin fraction accounts for 20–25% of wheat proteins and mainly includes protective and metabolic active proteins, such as enzymes and enzyme inhibitors [2]. Predominant in this protein fraction are alpha-amylase/trypsin inhibitors and purothionins which may have dual roles, as nutrient reserves for the germinating embryo and as inhibitors of insects and fungal pathogens prior to germination. It is assumed that albumins and globulins tend to be insensitive to applied fertilizer [4]. For spring wheat, a high baking quality is expected regardless of the grain yield. Studies on spring wheat cultivar responses to late-season $^{15}N$ application in terms of grain quality traits are limited. The scientific objective of this experiment was to quantify the amount of fertilizer nitrogen effectively used for protein synthesis in gluten and how this is affected by water deficit during the grain filling phase. To assess the effect of a late N fertilization at the onset of the heading amount of 30 kg N ha$^{-1}$ or even double the amount of 60 kg N ha$^{-1}$, known as quality effective application, two classified German spring wheat cultivars and two cultivars from the Mediterranean territories, Golia and Gönen, which are not classified, were examined. For this, the $^{15}N$ dilution technique was used to discriminate the N that is absorbed during the grain filling phase. This fertilization was combined with a reduced water supply, resulting in ~11 vol% compared to ~18 vol% of the well-watered plants. Beginning at the phenological stage, inflorescence fully emerged, for 21 days, and the effect on gluten and non-gluten proteins was examined. Summarized, under semi-controlled conditions in a pot experiment, the experimental members were (a) four cultivars (Taifun, Monsun, Golia, Gönen), (b) two different N applications at heading (30 kg or 60 kg N ha$^{-1}$), and (c) two water treatments (well-watered, and water deficit) with the beginning end of the heading with three replications each.

## 2. Materials and Methods

### 2.1. Plant Material

Classified German spring wheat cultivars (*Triticum sativum* L.), Elite wheat (E) Taifun (KWS SAAT SE & Co. KGaA), Quality wheat class A, Monsun (KWS SAAT SE & Co. KGaA), and two cultivars from the Mediterranean territory, released in Turkey (1989, 1989, respectively) Golia an alternative type and Gönen, a Turkish spring type, both grown in Turkey, but both not classified (nc), were selected. A randomized pot experiment was conducted in an open wire house with a Perspex rain shelter. Five-liter Mitscherlich pots were filled with 6.5 kg soil (Albic Luvisol, poor silty to medium loamy sand), pH 6.0, with 0.11 % N, 1.22 % C, 400.4 mg P kg$^{-1}$, 279.4 mg K kg$^{-1}$. Fifteen seeds were sown per pot in each of three replications, at emergence thinned to ten plants, and at onset of stem elongation to eight plants, and supplied with 82 mg P and 313 mg K pot$^{-1}$.

### 2.2. Fertilization and Water Supply

Three splits of N-fertilizer were applied as $NH_4NO_3$ in a 200 mL solution at emergence, the onset of stem elongation, and the beginning of heading; the tip of inflorescence emerged from the sheath first spikelet just visible (EC 10, 30, and 51, respectively [5]). Rates (mg N pot$^{-1}$) were 283, 141, 94 or 188, representative for 90 + 45 + 30 or 60 kg N ha$^{-1}$. Ten percent of the third N-split (9.4, 18.8 mg $^{15}N$, respectively) was applied as $^{15}NH_4NO_3$ (96.4 atom %, Campro Scientific GmbH, Germany). In the well-watered (ww) plants, soil moisture was maintained at ~18 vol%. From the phenological stage, the end of heading: inflorescence fully emerged (EC 59), the soil moisture of the water deficit (wd) plants was kept at ~11 vol% for 21 days. Afterwards, these plants were rewatered to the level of the well-watered plants, and all plants were allowed to mature. The water regime during the experimental period was monitored by Time Domain Reflectometry (TDR). In one pot of each of the experimental members, one TDR-probe was vertically installed in the middle of the pot. The water content of the soil was monitored every three days

(Monday, Wednesday, Saturday) beginning at about 10 a.m. The maximum soil moisture was ~20 vol%, whereby ~18 vol% was considered as optimum for the well-watered plants, and ~11 vol% for the water deficit plants. The difference between water and the target value was calculated by 'the mathematical rule of three' and afterward applied. Cultivars were according to the demand differentiated supplied with water. Altogether, between first leaf through coleoptile (EC 10) and physiological maturity (EC 89), the wd plants received ~9% less water compared to the ww-plants. Plants were harvested at the fully ripe stage, grain hard (EC 89), dried at 60 °C to constant weight, and kernels were obtained by using an ear thresher, Walter-Wintersteiger, Obernberg/Inn Austria. All experimental measures/applications during the pot experiment shown in Table S1.

### 2.3. Protein Isolation

The NaCl insoluble storage (gluten) and soluble proteins (non-gluten) were prepared according to ICC Standard 106/2 (http://www.icc.or.at/methods, accessed on 1 January 2017), adapted by isolation of 4 g whole grain flour (0.5-mm screen, Hammer Type Laboratory Mill 120, Perten Instruments) with 4 mL 2% NaCl solution in a porcelain dish, filtrates above 'Müller gaze' (pore size 360 μm) and washed 4 times with 2 mL 2% NaCl solution. Thereafter, these samples were dried and the protein content (N × 6.25) was determined after Kjeldahl nitrogen analysis. Measurement of $^{15}$N in gluten was accomplished by isotope ratio mass spectrometry (IRMS) with the Tracer Mass 20–20, SerCon, Crewe, UK, and the corresponding $^{15}$N content was calculated [6].

### 2.4. Statistical Analysis

The data (mean, standard error (SE), ANOVA, Student's *t*-test, univariate model) were analyzed using statistical software IBM SPSS Statistic 25.0. ANOVA stands for analysis of variance and is used to compare the means of more than 2 groups. It is an extension of the Student's *t*-test which compares the mean values of a maximum of 2 groups. Within statistics, univariate expresses that the measured variable under consideration is one-dimensional, even if it depends on several variables. This is particularly the case when the measured variable is the one-dimensional dependent variable of a random experiment or the characteristic value of a one-dimensional random variable.

### 3. Results and Discussion

The vegetation time is defined as seeding to maturity (March to July 2017); for Taifun, Monsun, Golia, and Gönen, the duration was 119, 120, 125, and 132 days, respectively. The stage beginning of heading, tip of inflorescence emerged from the sheath, first spikelet was just visible (EC 51), – the time point of the 3rd 15N/N split –, was at the day of year (DOY) 141, 143, 147, and 153. The duration from EC 51 to the phenological stage, end of heading: inflorescence fully emerged (EC 59), was in the order mentioned with 5, 4, 3, and 2 days, and relatively short. From this stage on, the soil moisture of the well-watered plants was maintained at ~18 vol%, whereas the restrictive water supply for 21 days of the wd plants began. The phenological phase from EC 59 to EC 89, fully ripe, was similar for Taifun, Monsun, Golia, and Gönen and lasted 58, 57, 58, and 59 days, respectively, representing on average ~40% of the vegetation time. The duration of this phase was facing the time from the beginning of the linear protein accumulation to cessation of protein accumulation, of 10- and 36-days post anthesis [4].

For the data of the protein content in gluten, shown in Table 1, the univariate analysis for cultivars, the treatment of well-watered plants, water deficit plant, and for the interaction of cultivar * fertilizer N (30 or 60 kg N) and the interaction of cultivar * treatment (well-watered, water deficit) reveals significant differences ($p < 0.001$, $p < 0.001$, $p < 0.027$, $p < 0.017$).

**Table 1.** Protein content in gluten (mean ± SE mg protein/g DM) at maturity of spring wheat cultivars influenced by N fertilization of 30 or 60 kg N applied at beginning of heading (EC 51) of well-watered and water deficit plants, with the beginning at the end of heading (EC 59) for 21 days (for each N level: well-watered plants: different capitals indicate significant differences between cultivars, Tukey HSD test, $p = 0.05$, $n = 3$; water deficit plants: different small letter indicates significant differences between cultivars, Tukey HSD test, $p = 0.05$, $n = 3$, * indicates significant difference between well-watered and water deficit plants, Student's *t*-test).

| Cultivar | Fertilizer N (kg) | Treatment | Gluten (mg Protein/g DM) | Fertilizer N (kg) | Treatment | Gluten (mg Protein/g DM) |
|---|---|---|---|---|---|---|
| Taifun | 30 | well-watered | 230.9 ± 7.2 A | 60 | well-watered | 242.7 ± 7.7 A |
| | 30 | water deficit | 248.9 ± 10.9 b | 60 | water deficit | 253.4 ± 9.6 b |
| Monsun | 30 | well-watered | 193.8 ± 7.3 B | 60 | well-watered | 212.7 ± 5.2 A |
| | 30 | water deficit | 202.3 ± 8.98 b | 60 | water deficit | 224.0 ± 8.7 b |
| Golia | 30 | well-watered | 245.0 ± 8.0A | 60 | well-watered | 230.4 ± 12.9 A |
| | 30 | water deficit | 296.7 * ± 16.6 a | 60 | water deficit | 276.5 * ± 7.7 a |
| Gönen | 30 | well-watered | 247.0 ± 4.3 A | 60 | well-watered | 243.9 ± 2.7 A |
| | 30 | water deficit | 256.6 ± 5.8 a | 60 | water deficit | 287.4 * ± 7.8 a |

The protein content in gluten (Table 1, mg protein/g DM) was for the N fertilization of 30 kg, applied at the beginning of heading stages and well-watered, was similar for Taifun, Golia, and Gönen, but markedly lower for the quality wheat Monsun, 231, 245, 247, and 194 mg protein/g DM, respectively. Water deficit tends to increase the protein content in gluten, whereby the protein content of the two Mediterranean cultivars, Golia and Gönen, was significantly higher than that of Taifun and Monsun 297, 257, 249, and 202 mg protein/g DM, respectively. Doubling the amount of fertilizer N to 60 kg under well-watered conditions, did not result in any significant change of the protein content in gluten and differences between the cultivars, 243, 213, 230, 244 mg protein/g DM respectively, were not detectable. As shown for 30 kg N, combined with the water deficit for Gönen, the protein content in gluten of the two Mediterranean cultivars, Golia and Gönen, was significantly higher than that of Taifun and Monsun 277, 287, 253, and 224 mg protein/g DM, when 60 kg N was applied. A comparison between the well-watered and the water deficit plants showed for Golia for the N application of 30 kg N, and for 60 kg N for both the Mediterranean cultivars, Golia and Gönen, an ~19% markedly (* $p = 0.05$) rising protein content in gluten, 297, 277, and 287 mg protein/g DM, compared to the well-watered plants (Table 1).

The pre-anthesis, 'source' development time frame, assimilation, accumulation, and translocation of nitrogenous compounds is followed by the period from anthesis to grain maturity, when the growth and potential size of the grains act as 'sinks' for N. It is well known that the N requirement for protein synthesis in the developing wheat kernel is met by 50–70% of the mobilization of previously-assimilated N present in vegetative tissues, as leaves, stems, glumes, and also by direct uptake and assimilation of N during grain filling. The main source of proteins stored in wheat kernels is remobilized leaf proteins as Rubisco (ribulose bisphosphate carboxylase/oxygenase) [7–9]. It is also known that the remobilization and re-allocation ability, however, can differ among wheat cultivars, e.g., from the Mediterranean territory and was for Golia about 10% higher as in Gönen [10]. The late [15]N application of 45 kg N at the beginning of heading (EC 51) under well-watered conditions led to partitioning into grains of 89%; i.e., the third N-split was most effectively transferred into the grains, although values did not differ between Golia and Gönen [10].

The biosynthesis of cereal grain storage proteins (GSP) is under developmental and nutritional regulation, and they are specifically synthesized in the endosperm where they accumulate. The dynamics of metabolite pools, either transported by the phloem or synthesized de novo in developing grains may play a key role in the signal transduction between nutrient availability and grain storage protein accumulation [11]. Measurements of GSP, targeted metabolites, and transcript contents under a variety of N conditions

showed that allometric allocation of N to GSPs was regulated at the transcriptional level and that several transcription factors (e.g., HMG, AP2-EREBP, MYBS3, FUSCA3, and MCB1) were involved. Furthermore, changes in the expression of genes involved in transport and metabolism modulated the concentrations of free amino acids in response to plant nutritional status, leading to altered GSP accumulation [3] and references therein. Moreover, Barneix [12] summarized a number of possible regulatory 'switching points', including nitrate uptake by roots, nitrate reduction within the root tissue, capacity of protein synthesis in the chloroplasts, protein hydrolysis, amino acid export to the phloem, and also the potential for protein synthesis in the grain. The specific factors and/or their interactions leading to a lower protein content in the gluten of Monsun, compared to the other three cultivars, were not possible to evaluate. It seems that the amount of late application of 30 kg N is limiting the gluten synthesis in grains of Monsun, because when 60 kg N were applied, no differences occurred between cultivars (Table 1). Compared to the [15]N uptake via the soil, for example, [15]N-urea applied by foliar application in a field experiment at flag leaf sheath opening, was after 35 days at maturity to 16% involved in the gluten synthesis, and the [15]N content was not statistically different between Taifun, Monsun, Golia, and Gönen [13].

For the data of the protein content of non-gluten proteins (Table 2), the univariate analysis for cultivars and for the interaction cultivar * treatment (well-watered, water deficit) reveals significant differences ($p < 0.001$, $p < 0.006$).

**Table 2.** Protein content of non-gluten (mean $\pm$ SE mg protein/g DM) at maturity of spring wheat cultivars influenced by N fertilization of 30 or 60 kg N applied at the beginning of heading (EC 51) of well-watered and water deficit plants, with the beginning at the end of heading (EC 59) for 21 days. (for each N level: well-watered plants: different capitals indicate significant differences between cultivars, Tukey HSD test, $p = 0.05$, $n = 3$; water deficit plants: different small letter indicates significant differences between cultivars, Tukey HSD test, $p = 0.05$, $n = 3$).

| Cultivar | Fertilizer N (kg) | Treatment | Non-Gluten (mg Protein/g DM) | Fertilizer N (kg) | Treatment | Non-Gluten (mg Protein/g DM) |
|---|---|---|---|---|---|---|
| Taifun | 30 | well-watered | 48.8 $\pm$ 3.2 A | 60 | well-watered | 48.6 $\pm$ 0.9 A |
| | 30 | water deficit | 45.9 $\pm$ 0.7 b | 60 | water deficit | 40.2 $\pm$ 5.3 a |
| Monsun | 30 | well-watered | 46.9 $\pm$ 0.7 A | 60 | well-watered | 45.3 $\pm$ 1.0 A |
| | 30 | water deficit | 45.5 $\pm$ 1.0 b | 60 | water deficit | 50.4 $\pm$ 1.1 a |
| Golia | 30 | well-watered | 47.1 $\pm$ 1.4 A | 60 | well-watered | 47.2 $\pm$ 1.2 A |
| | 30 | water deficit | 47.1 $\pm$ 1.4 b | 60 | water deficit | 45.8 $\pm$ 0.9 a |
| Gönen | 30 | well-watered | 48.6 $\pm$ 1.3 A | 60 | well-watered | 49.6 $\pm$ 3.6 A |
| | 30 | water deficit | 56.0 $\pm$ 1.7 a | 60 | water deficit | 56.2 $\pm$ 4.7 a |

The content of non-gluten proteins (Table 2) for the well-watered plants was not different between the cultivars and not influenced by the N fertilization of 30 or 60 kg N, 49, 47, 47, 49, and 49, 45, 47, 50 mg protein/g DM, respectively. Non-gluten proteins mainly consist of globulins and albumins which predominantly accumulate during the early phase of grain growth, when endosperm cells are still dividing, whereas the accumulation of storage or gluten proteins occurs later when cell division has stopped and cell body growth is due only to cell expansion [14]. Although for total albumin and globulin, the non-gluten proteins tend to be insensitive to applied fertilizer [4]. This could also be confirmed here, but also different seeding rates, combined with or without additional water supply under field conditions [15] does not change the [15]N content of non-gluten proteins. Studies on the regulation of albumin and globulin genes in response to the environment have until now received only little attention. Moreover, the growing location and maturation conditions are the origin of the differences in distribution among the gluten storage protein glutenin polymers and in the mode in glutenin polymerization. It could be shown [16] that grains of winter wheat from a conventional production system (four N applications: 60 + 40 + 20 + 20 = 180 kg N ha$^{-1}$) compared to an integrated cropping system (three

N applications: 40 + 30 + 20 = 90 kg N ha$^{-1}$) contained significantly more gliadins and glutenins, thus gluten, and a dry wheat maturing period was favorable for a significantly higher value in gluten. Yet, the content of the sum of albumins + globulins was under these conditions only marginally influenced by the different cropping systems.

For the data of the $^{15}$N protein content in gluten (Table 3), the univariate analysis for the cultivars, for the fertilizer N (30 or 60 kg N), the treatment well-watered, water deficit, and for their interactions cultivar\*fertilizer N (30 or 60 kg N), cultivar \* treatment (well-watered, water deficit), fertilizer N (30 or 60 kg N) \* treatment (well-watered, water deficit) reveals significant differences ($p < 0.0001$, $p < 0.0001$, $p < 0.0001$, $p < 0.006$, $p < 0.0001$, $p < 0.001$).

**Table 3.** $^{15}$N protein in gluten (mean $\pm$ SE mg protein/g DM) at maturity of spring wheat cultivars influenced by N fertilization of 30 or 60 kg N applied at beginning of heading (EC 51) of well-watered and water deficit plants, with the beginning at the end of heading (EC 59) for 21 days. (for each N level: well-watered plants: different capitals indicate significant differences between cultivars, Tukey HSD test, $p = 0.05$, $n = 3$; water deficit plants: different small letter indicates significant differences between cultivars, Tukey HSD test, $p = 0.05$, $n = 3$; \* indicate significant difference between well-watered and water deficit, Student's *t*-test).

| Cultivar | Fertilizer N (kg) | Treatment | Gluten (mg $^{15}$N Protein/g DM) | Fertilizer N (kg) | Treatment | Gluten (mg $^{15}$N Protein/g DM) |
|---|---|---|---|---|---|---|
| Taifun | 30 | well-watered | 1.64 $\pm$ 0.1 C | 60 | well-watered | 3.50 $\pm$ 0.1 AB |
| | 30 | water deficit | 1.81 $\pm$ 0.1 c | 60 | water deficit | 3.62 $\pm$ 0.1 c |
| Monsun | 30 | well-watered | 1.50 $\pm$ 0.1 BC | 60 | well-watered | 3.21 $\pm$ 0.2 B |
| | 30 | water deficit | 1.69 $\pm$ 0.1 c | 60 | water deficit | 3.50 $\pm$ 0.2 c |
| Golia | 30 | well-watered | 1.93 $\pm$ 0.1 B | 60 | well-watered | 3.50 $\pm$ 0.2 AB |
| | 30 | water deficit | 2.38 \* $\pm$ 0.1 b | 60 | water deficit | 4.34 \* $\pm$ 0.1 b |
| Gönen | 30 | well-watered | 2.32 $\pm$ 0.1 A | 60 | well-watered | 4.02 $\pm$ 0.1 A |
| | 30 | water deficit | 2.99 \* $\pm$ 0.1 a | 60 | water deficit | 5.87 \* $\pm$ 0.1 a |

The third N split of 30 or 60 kg N was $^{15}$N-labeled and the results for the $^{15}$N gluten protein (Table 3) show a more differentiated picture compared to the unlabeled gluten protein (Table 1). The $^{15}$N protein in gluten (Table 3, mg $^{15}$N protein/g DM) for the N fertilization of 30 kg, applied at the beginning of heading stages and well-watered, was markedly higher in Gönen with 2.32 mg $^{15}$N protein/g DM, compared to Golia and Monsun, which contained 1.93 and 1.50 mg $^{15}$N protein/g DM, respectively, and similar in Taifun with 1.64 mg $^{15}$N protein/g DM. Remarkably, $^{15}$N fertilizer uptake into gluten is stimulated by water deficit, at a soil moisture of ~11 vol%, at 30 and 60 kg N, and leads to significantly increased $^{15}$N levels in both Golia and Gönen containing 2.38, 2.99, 4.34, and 5.87 mg $^{15}$N protein/g DM, respectively, at 23%, 29%, 24%, and 46%. Doubling the dose of fertilizer N from 30 to 60 kg led to a proportional two-time increase in the $^{15}$N protein/g DM in the gluten of the four cultivars, both in well-watered and water deficit plants. For these cultivars, independent of classification and origin, this is a clear indication of a potential for utilizing a fertilizer N amount of minimum 60 kg N for the protein synthesis of storage proteins. It was shown [3] that moderately 'high' N fertilization induces the expression of transcription factors with N-responsive elements (GRF4, MYBS3), which accelerates the uptake and assimilation of ammonium, which was $^{15}$N labelled in this experiment. Proteins or enzymes such as phosphoglycerate mutase (PGM) and malate dehydrogenase (MDH) involved in protein synthesis are phosphorylated under high-N conditions, increasing their activities and accelerating storage protein synthesis and accumulation. These synergistic effects result in the upregulation of storage protein components and increased gliadin and glutenin content, ultimately improving dough viscoelasticity and breadmaking quality.

The distribution patterns and concentration gradient of the protein and protein components in wheat grain were investigated using the pearling technique, combined with the $^{15}$N isotope tracing technique, used to discriminate the N sources of different protein components in different pearling fractions of grain [9]. The quantitative and qualitative gra-

dients of the protein distribution appear to follow the radial pattern of cell development in the endosperm. Gradients in the gluten protein composition are related to the origin of the subaleurone cells, which are different from other starchy endosperm cells that derived from the re-differentiation of aleurone cells, but it could also be related to the signals produced by the maternal tissue that affect specific domains of the gluten protein gene promoters.

## 4. Conclusions

It can be concluded that such results will be essential in understanding the mechanisms of grain quality formation and developing better N fertilization protocols. Adapting the N application rate during the different phenological phases and considering cultivar specific requirements should be an effective approach to regulating the distribution of the protein fractions in the grains for specific end use, especially under water limited conditions. To achieve this goal, the $^{15}$N dilution technique is a very effective method, under both standardized and field conditions.

**Supplementary Materials:** The following supporting information can be downloaded at: https://www.mdpi.com/article/10.3390/nitrogen3040041/s1, Table S1: Experimental measures/applications during the pot experiment.

**Author Contributions:** K.-P.G. and O.E. contributed equally to writing, reviewing, and editing the paper. All authors have read and agreed to the published version of the manuscript.

**Funding:** This research received no external funding.

**Data Availability Statement:** Not applicable.

**Conflicts of Interest:** The authors declare no conflict of interest.

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
