# Peer review of "Effect of Late 15N-Fertilization and Water Deficit on Allocation into the Gluten of German and Mediterranean Spring Wheat Cultivars"

_nitrogen, doi:10.3390/nitrogen3040041_

Round 1
Reviewer 1 Report
I amcomfortable with the paper as it is presently prepared. Represents a significant contribution to the subject researched.
Author Response
Reviewer 1
We are very thankful for reviewing our manuscript carefully!

Reviewer 2 Report
The manuscript is well-written and informative, but there are a few points that need to correct before publishing.
in section "2.2. Fertilization and water supply" the authors should rewrite this section with more details and explanation to allow the reader to understand the research purpose.
3. Results and Discussion ....line 109 "March to July" which year?
line 172 [3 and references therein]" what do you mean by references therein?
I think the authors need to write a conclusion
Author Response
We appreciate the constructive and helpful comments that have helped improve the MS!
Reviewer 2
Comment: in section "2.2. Fertilization and water supply" the authors should rewrite this section with more details and explanation to allow the reader to understand the research purpose.
Response: Revised. The content of this paragraph is extended; see line 99-105.
Comment: 3. Results and Discussion ....line 109 "March to July" which year?
Response: Revised. 2017; see line 133.
Comment: line 172 184 [3 and references therein]" what do you mean by references therein?
Response: In this paragraph, results are quoted directly from Zhen, but also two relevant ones that were given by Zhen. This way of citation, if it is not predominantly practiced, is quite recognized in scientific articles.
line 194-200:
Measurements of GSP, targeted metabolites, and transcript contents under a variety of N conditions showed that allometric allocation of N to GSPs was regulated at the transcriptional level and that several transcription factors (e.g., HMG, AP2-EREBP, MYBS3, FUSCA3, and MCB1) were involved. Furthermore, changes in the expression of genes involved in transport and metabolism modulated the concentrations of free amino acids in response to plant nutritional status, leading to altered GSP accumulation [3 and references therein].
Comment: I think the authors need to write a conclusion
Response: Revised; see line 294-301.

Reviewer 3 Report
Research is relevant and interesting.
1. No research hypothesis is presented in the introduction;
2. I recommend presenting the discussion in a separate chapter. The discussion with other authors could have been more detailed;
3. Methods of statistical analysis need to be described more clearly;
4. The Latin name of spring wheat is not specified;
5. The years of the investigations are not indicated;
6. I recommend agrotechnologies of the experiment to present in the table;
7. Few literature sources are analyzed in the article;
8. I recommend presenting the conclusions in a separate chapter;
9. I suggest providing photos of the experiment.
Author Response
We appreciate the constructive and helpful comments that have helped improve the MS!
Reviewer 3
- No research hypothesis is presented in the introduction
Response: Revised. In this case a “scientific objective” was formulated (line 59-66).
- I recommend presenting the discussion in a separate chapter.The discussion with other
authors could have been more detailed
Response: The aim of this article (7 pages) was to summarize the results of the one-year pot experiment, for 14N, for Gluten and Non-Gluten; and for 15N for Gluten during grain filling phase. The scope of the results (Tab. 1-3) can therefore be discussed well in one chapter (R & D). It should be considered that there is little literature on the topic dealt with in this paper.
- Methods of statistical analysis need to be described more clearly
Response: Revised; see line 124-130.
- The Latin name of spring wheat is not specified
Response: Revised; (Triticum sativum L.) see line 177
- The years of the investigations are not indicated
Response: Revised 2017; see line 133.
- I recommend agrotechnologies of the experiment to present in the table
Response: An overview (Table) of the experimental setup is included as supplementary material (S1).
- Few literature sources are analyzed in the article
Response: see line 59-60.
- I recommend presenting the conclusions in a separate chapter
Response: Revised; see line 294-301.
- I suggest providing photos of the experiment
Response: For illustration, no photos available.

Round 2
Reviewer 3 Report
No comments.